# Efficacy of the Yeast *Wickerhamomyces anomalus* in Biocontrol of Gray Mold Decay of Tomatoes and Study of the Mechanisms Involved

**DOI:** 10.3390/foods11050720

**Published:** 2022-02-28

**Authors:** Boen Lanhuang, Qiya Yang, Esa Abiso Godana, Hongyin Zhang

**Affiliations:** Schoold of Food and Biological Engineering, Jiangsu University, Zhenjiang 212013, China; lh_bourn@163.com (B.L.); yangqiya1118@163.com (Q.Y.); esa.abiso@hotmail.com (E.A.G.)

**Keywords:** tomato, biocontrol, *Wickerhamomyces anomalus*, enzyme activity, *Botrytis cinerea*

## Abstract

Gray mold decay is a widespread postharvest disease in tomato that results from infection by the pathogen *Botrytis cinerea*, leading to huge economic losses. The objective of this study was to select the most effective antagonistic yeast to control tomato gray mold from six potential biocontrol agents and to investigate the possible control mechanism. The results showed that the yeast *Wickerhamomyces anomalus* was the most effective in inhibiting *B. cinerea* among the six strains both in vivo and in vitro on tomato, with a colony diameter of 11 mm, a decay diameter of 20 mm, and the lowest decay incidence (53%)—values significantly smaller and lower than the values recorded for the control group and the other yeasts. The efficacy of the control depended on the increase in yeast concentration, and the decay incidence and lesion diameter were reduced to 31%, 28% and 7 mm, 6 mm, respectively, when treated with 1 × 10^8^ and 1 × 10^9^ cells/mL *W. anomalus*. In addition, *W. anomalus* was able to rapidly colonize and stably multiply in tomato, occupying the space to control pathogen infection. *W. anomalus* was also able to motivate the defense mechanism of tomato with stimulation of defense-related enzymes PPO, POD, APX, and SOD and promotion of the content of total phenols and flavonoid compounds. All these results suggest that *W. anomalus* exhibited exceptional ability to control gray mold in tomato.

## 1. Introduction

Tomato (*Solanum lycopersicum*) is a widely grown and consumed horticultural crop with high economic and nutritional value. However, the tomato industry suffers great losses due to pre- and postharvest infection by pathogens, especially *Alternaria solani* [1] and *B. cinerea* [2], which cause early blight and gray mold, respectively, and lead to fruit quality deterioration during harvest, transportation, and storage, negatively affecting crop yield and shelf life. Gray mold has been reported to be one of the most widespread diseases in greenhouse tomatoes, leading to a sharp increase in decay incidence and severe yield losses [3]. *B. cinerea* infection originates in aged or injured tissues of fruits, causes discoloration and postharvest rot, and spreads rapidly to plant leaves, stems, and fruits, which also leads to severe deterioration of tomatoes during storage and shelf life [4,5].

Chemical control has always been considered an economical method with high efficacy to control tomato gray mold, and some fungicides such as carbendazim, diethofencarb, and procymidone have been used in fields for decades [6,7]. However, researchers found that strong resistance against many fungicides has been developed by *B. cinerea*, leading to an increase in the dose or even intensive use of these chemicals in fields and greenhouses [8]. Moreover, the excessive use of chemical fungicides leads to chemical residues and environmental damage that endanger the health of humans and other living organisms [9,10]. Therefore, there is an urgent need to shift from this traditional method to a new, safer method.

Biological control could be an alternative to protect fruits and vegetables from postharvest diseases. Compared to chemical methods, biological control does little harm to the environment and humans through the use of antagonistic microorganisms, especially yeasts, and carries a lower risk of pathogen resistance development due to complex interaction mechanisms [11]. To investigate the possibility of reducing postharvest gray mold with this method, many potential biocontrol agents and their biocontrol mechanisms have been explored. It was discovered that *Rhodosporidium paludigenum* was able to effectively inhibit gray mold in cherry tomatoes via nutritional competition [12]. Guo et al. reported that volatile compounds of *Hanseniaspora uvarum* played an important role in inhibiting the mycelial germination of *B. cinerea* and controlling decay in cherry tomatoes [13]. Among the numerous biocontrol agents, *W. anomalus* showed high efficacy against *B. cinerea* by inhibiting spore germination in vitro and curbing the development of gray mold in tomato plants [14]. *W. anomalus* is an antagonistic yeast for which our research team recently reported the efficacy of its biocontrol activity against *Penicillium expansum* infection on pears, with no acute toxicity hazard [15]. Although many biocontrol agents were reported to effectively reduce decay in crops and although the use of *W. anomalus* for biocontrol of gray mold in tomato has been attempted, further research into the possible biocontrol mechanisms is still needed.

This study aimed to: (1) evaluate and compare the biocontrol efficacy of six antagonistic yeasts on *B. cinerea*; (2) evaluate the efficacy of *W. anomalus* with a concentration gradient against gray mold in tomato; (3) determine the colonization trends of *W. anomalus* in tomato tissue and epidermis at 4 °C and 20 °C; and (4) evaluate the possible mechanism involved in biocontrol of gray mold in tomato.

## 2. Materials and Methods

### 2.1. Fruits

Tomatoes (cv. Zhefen 301) were harvested at commercial maturity (Late Red Ripening, covered with 70−100% red, according to standard GH/T 1193-2021) in an orchard in Zhenjiang, Jiangsu Province, China with a uniform size (250 ± 20 g in weight and 65 ± 5 mm in diameter) and showed no physical damage or disease. All fruits were cleaned with tap water and soaked in sodium hypochlorite (3% *v*/*v*) for 2 min, then rinsed and placed in sterile plastic baskets for natural drying.

### 2.2. Antagonist

The antagonistic yeasts were previously isolated and identified by constructing a phylogenetic tree using *ITS* domain sequences and stored in an ultracold freezer (−80 °C) containing 25% glycerol. *Pichia manshurica* was isolated from a vineyard of Jvrong, Zhenjiang, Jiangsu Province and deposited in the China General Microbiological Culture Collection Center (CGMCC), under the number CGMCC 2.5415; *Cryptococcus laurentii* was isolated from a vineyard of Jvrong, Zhenjiang, Jiangsu Province and stored at −80 °C; *Sporidiobolus pararoseus* was isolated from a vineyard of Shangdang, Zhenjiang, Jiangsu Province and deposited in the CGMCC, under the number CGMCC 2.5351; *Saccharomyces cerevisiae* was isolated from Shiyezhou Agro Ecological Orchard, Zhenjiang, Jiangsu Province and deposited in the CGMCC, under the number CGMCC 7129; *Pichia caribbica* was isolated from Jiangxinzhou Agro Ecological Orchard, Zhenjiang, Jiangsu Province and deposited in the CGMCC, under the number CGMCC 3616; and *W. anomalus* was isolated from orchard soil in Zhenjiang, Jiangsu Province and deposited in the China Center for Type Culture Collection (CCTCC), under the number CCTCC M 2018053 [15]. Then, 10 μL of stored yeast liquid was inoculated into 50 mL of nutritional yeast dextrose broth (NYDB) and mixed at a constant temperature (180 rpm, 28 °C) for 24 h, and the second activation was performed with a 1% inoculation amount under the same conditions. The yeast cells were collected, purified, and suspended with normal saline according to the methods of He et al. [16]. Then the yeast solution was diluted to the required concentration (1 × 10^8^ cells/mL for experiments conducted under Section 2.4, Section 2.6, and Section 2.7 and 1 × 10^6^, 1 × 10^7^, 1 × 10^8^, 1 × 10^9^ cells/mL for Section 2.5) after counting with a hemocytometer.

### 2.3. Fungal Pathogen

The pathogen *B. cinerea* was separated and purified from decayed tomatoes, identified by comparing sequence similarity with records in the NCBI database and preserved at −80 °C. Then, 200 μL of the stored solution was mixed with 50 mL potato dextrose broth (PDB) and incubated at 25 °C with 180 rpm for 24 h. The activated fungi were incubated in potato dextrose agar (PDA) using the spread plate method and cultured in an incubator at constant temperature (25 °C) for 7 days. Spore suspension of the pathogen was prepared by scraping the spores from the plate surfaces and counting them with a hemocytometer and adjusting them to the required concentrations with normal saline.

### 2.4. Inhibitory Activity of Different Yeasts against Tomato Gray Mold Caused by B. cinerea

In vitro, a 6 mm diameter hole was pierced in each solidified PDA culture in a Petri dish using a sterilized hole punch and 15 μL of (1) 1 × 10^8^ cells/mL yeast solution as treatment (Y1: *P. manshurica*; Y2: *C. flavescens*; Y3: *S. pararoseus*; Y4: *S. cerevisiae*; Y5: *P. caribbica*; Y6: *W. anomalus*) and (2) sterilized normal saline as control (CK) were inoculated into each hole. After 2 h, 15 μL *B. cinerea* suspension at a concentration of 1 × 10^5^ spores/mL was also inoculated. After 4 d in the incubators (25 °C, 95% RH), the colony diameter of each Petri dish was measured. Each treatment included three repeats of five Petri dishes and the experiment was conducted twice.

For the in vivo experiment, three uniform wounds (3.5 mm diameter, 5 mm depth) were made in the centre of each tomato by piercing with a sterile punch, then 15 μL 1 × 10^5^ spores/mL *B. cinerea* suspension was inoculated into each wound after 15 μL (1) 1 × 10^8^ cells/mL yeast solution, (2) sterilized normal saline. All tomatoes were placed in clean plastic baskets, wrapped with cling film, and placed in the incubators (20 °C, 95% RH). After four days, decay incidence and diameter were recorded and measured. Each treatment includes three replicates of 12 tomatoes and the experiment was conducted two times.

### 2.5. Efficacy of W. anomalus at Different Concentrations against Tomato Gray Mold Caused by B. cinerea

Three equal wounds were made in each tomato and 15 μL of (1) *W. anomalus* solution at different concentrations (1 × 10^6^, 1 × 10^7^, 1 × 10^8^, 1 × 10^9^ cells/mL) and (2) sterilized normal saline (CK) was inoculated into each wound, and inoculation of 15 μL of *B. cinerea* suspension (1 × 10^5^ spores/mL) was carried out after 2 h. After four days of storage (20 °C, 95% RH), the decay incidence was recorded and lesion diameter was measured. Each treatment included three replicates of 12 tomatoes and the experiment was conducted twice.

### 2.6. Colonization Trends of W. anomalus in Tomato Tissue and Epidermis at 20 °C and 4 °C

As described above, 15 μL *W. anomalus* (1 × 10^8^ cells/mL) were inoculated into wounds or applied to the surface (within a circle, each 30 mm in diameter) of cleaned tomatoes. All tomatoes were placed in cleaned baskets with cling film and divided into two groups, which were stored at 20 °C and 4 °C, respectively. A sterile scalpel was used to collect samples of the tomato wounds and epidermis. At 20 °C, tomato wounds were sampled daily from 0 to 7 days and epidermis was sampled every 12 h until 144 h; at 4 °C, samples were collected every 3 days for 21 d. Samples were crushed using a tissue grinder (60 Hz, 7 min) and suspended with 30 mL of normal saline and applied to nutritional yeast dextrose agar (NYDA) after gradient dilution. After incubation at 28 °C for 2 d, the colonies of *W. anomalus* were counted and the results were transferred into lg CFU/wound and lg CFU/circle. Each treatment included three replicates and the experiment was carried out twice.

### 2.7. Efficacy of W. anomalus on Enzyme Activities and Content of Total Phenols and Flavonoid Compounds in Tomatoes

Tomatoes were cleaned, wounded, inoculated with (1) *W. anomalus* (1 × 10^8^ cells/mL) and (2) sterilized normal saline and stored as described above (20 °C, 95% RH). Tomato tissue (1.0 g) around each wound was collected on days 0, 1, 2, 3, 4, 5, 6, and 7 after treatment, ground (60 Hz, 7 min) with extraction buffer (Table 1) and centrifuged at 12,000 rpm (4°C) for 15 min. The supernatant was collected as extracted solution. Each treatment included three replications and the experiment was conducted twice.

The activity of the enzymes was determined according to the methods in Table 1, with some modifications. The SOD activity was measured with photochemical reduction of nitro blue tetrazolium chloride (NBT) in a light box producing uniform illumination intensity. The content of total phenols and flavonoid compounds was extracted with 1% (*v*/*v*) HCl–methanol solution and measured according to the method of Deng et al. with modifications [21]. Enzyme activities were expressed as U/g fresh weight and one unit of polyphenol oxidase (PPO), peroxidase (POD) was defined as an increase in one absorbance per minute of one gram of sample at 470 nm and 420 nm, respectively. One unit of ascorbate peroxidase (APX) was defined as the decrease of 0.01 per minute from one gram of sample at 290 nm and one unit of superoxide dismutase (SOD) was defined as the amount of enzyme causing 50% decrease in photochemical reduction of nitro blue tetrazolium chloride (NBT). The unit content of total phenols and flavonoid compounds was defined as OD_280_/g fresh weight and OD_325_/g fresh weight, respectively.

### 2.8. Statistical Analysis

Data normality was checked and then data analysis was conducted using IBM SPSS version 22 (IBM Corp., Armonk, NY, USA). One way analysis of variance was performed; different yeast treatments, different concentration of *W. anomalus*, and different time treatment were used as factors based on the experiment conducted. Duncan’s multiple range test was used to compare mean differences of data with more than two groups, whereas a *t*-test was used for data with only two groups. Data for percentage of decay incidence was transformed into arcsine square root values to normalize distribution before ANOVA; the percentage of decay incidence shown is untransformed data. Treatments were considered significant if the *p*-value was less than 0.05 (*p* < 0.05).

## 3. Results

### 3.1. Inhibitory Effect of Different Yeasts against Tomato Gray Mold Caused by B. cinerea

After four days of cultivation with different yeasts on PDA, the colony diameter of *B. cinerea* in all treatment groups was significantly smaller than that in the control group, which was 85 mm. Among the treatments, the colony diameter of *B. cinerea* treated with *Pichia manshurica* was the largest at 27 mm, while those treated with *P. caribbica* and *W. anomalus* were the smallest, 13 mm and 11 mm, respectively (Figure 1).

In vivo, three treatments showed no difference from CK in decay incidence, namely, *P. manshurica*, *P. caribbica*, and *Sporidiobolus pararoseus*, while the decay incidence of the tomatoes inoculated with *W. anomalus* (Y6) was only 53% (Figure 2A). When the decay diameter of CK increased to 43 mm, it was significantly lower in three treatments, *Saccharomyces cerevisiae*, *Cryptococcus flavescens,* and *W. anomalus*, with 26 mm, 22 mm and 20 mm, respectively (*p* < 0.001) (Figure 2B).

### 3.2. Efficacy of W. anomalus at Different Concentrations against the Tomato Gray Mold Caused by B. cinerea

After four days of storage, tomatoes treated with *W. anomalus* at a concentration of 10⁷, 10⁸, and 10⁹ cells/mL showed a significant decrease in gray mold compared to the control group. In tomatoes treated with 10⁸ and 10⁹ cells/mL yeast suspensions, the decay incidence was 30.56% and 27.78%, respectively, with no significant difference, while it was 92% in the control group (Figure 3A). Compared to the control group, all treatments significantly deceased the lesion diameter and there was a positive correlation between the decrease and yeast concentration. Treatments with *W. anomalus* concentrations of 10⁸ and 10⁹ cells/mL caused the lowest lesion diameter (6.65 and 5.94 mm, respectively) in tomato, almost a quarter of the diameter of the control group (27.81 mm) (*p* < 0.001) (Figure 3B).

### 3.3. Colonization Trends of W. anomalus in Tomato Tissue and Epidermis at 20 °C and 4 °C

Incubated at 20 °C, the population of *W. anomalus* inoculated into wounds of tomato showed a sharp increase from 1.62 × 10^6^ CFU/wound on the first day to 6.26 × 10^7^ CFU/wound on the second day, then gradually increased to 1.63 × 10^8^ CFU/wound on the fifth day, and then slightly decreased (Figure 4A). In contrast, the population of *W. anomalus* on the surface of tomato decreased from 5.40 × 10^5^ CFU/circle to 2.88 × 10^5^ CFU/circle in the first 12 h, followed by fluctuation between 3.74 × 10^5^ and 9.90 × 10^5^ CFU/circle up to 108 h, and then remained stable at 6.50 × 10^5^ CFU/circle in the last 36 h (*p* < 0.001) (Figure 4B).

In tomato wounds at 4 °C, the population of *W. anomalus* increased dramatically from 1.46 × 10^6^ CFU/wound to 3.73 × 10^7^ CFU/wound in the first 3 d. After a slight increase, it remained almost the same at 6.50 × 10^7^ CFU/wound between days 6 and 15, followed by a gentle increase (Figure 5A). On the tomato surface, the population of *W. anomalus* increased from 5.40 × 10^5^ CFU/circle on day 0 and peaked (2.98 × 10^6^ CFU/circle) on day 9, followed by a steady decline to 1.10 × 10^6^ CFU/circle on day 21 (*p* < 0.001) (Figure 5B).

### 3.4. Effects of W. anomalus on Enzyme Activities and Total Phenolic and Flavonoid Compounds Content

As shown in Figure 6A, after a considerable increase to 116 U/g FW on day 2, the PPO activity of CK remained almost the same in the last days, while the activity of the treatment group peaked at 164 U/g FW on day 2 (*t* = 4.162, *df* = 4, *p* = 0.014), followed by another peak at 196 U/g FW, which was almost double that of CK on day 5 (*t* = 5.123, *df* = 4, *p* = 0.007). The POD activity of tomato in the control group fluctuated from day 0 to day 5 between 94 U/g FW and 116 U/g FW and then remained constant. The activity of the treatment group also fluctuated during the first three days, then increased sharply and peaked at 219 U/g FW on day 5, which was more than twice that of CK (*t* = 8.041, *df* = 4, *p* = 0.001), followed by a gradual decline (Figure 6B). The APX activity of CK and the treatment group showed a similar trend. After increasing from about 273 U/g FW to 313 U/g FW and 400 U/g FW, both curves gradually decreased, with slight fluctuations, to 84 U/g FW and 209 U/g FW, respectively. Moreover, apart from day 0, the activity of APX in the treatment group was higher than that of CK (*p* < 0.030) (Figure 6C). The SOD activity of CK showed a gradual increase with fluctuations from 3.0 U/g FW to about 5.0 at day 4 and then remained stable. The activity of the treatment group increased on day 1 and then increased mildly from 4.7 U/g FW to 5.2 U/g FW until day 4. After decreasing to 4.3 U/g FW, it increased and peaked at 6.4 U/g FW on day 6 (*t* = 3.910, *df* = 4, *p* = 0.017), followed by a sharp decrease (Figure 6D).

The content of total phenols was similar in both CK and treatment groups on the first day, increasing from about 0.170 OD_280_/g FW to about 0.247 OD_280_/g FW. Thereafter, the content in the CK fluctuated for four days before it increased sharply to 0.328 OD_280_/g FW on the seventh day, while the content in the treatment group increased steadily during these days and reached 0.309 OD_280_/g FW at the end (Figure 6E). For flavonoid compound content, the curve of CK increased from 0.167 OD_325_/g FW to 0.297 OD_325_/g FW and then fluctuated until day 4, and after a gradual increase it remained constant at about 0.390 OD_325_/g FW after day 6. In the treatment group, there was a significant increase from 0.191 OD_325_/g FW to 0.401 OD_325_/g FW on day 4, which was 1.4 times the value of CK (*t* = 3.966, *df* = 4, *p* = 0.017), followed by a mild fluctuation (Figure 6F).

## 4. Discussion

In recent years, the ability of antagonistic yeasts to biologically control postharvest diseases of fruits and vegetables has been extensively studied. *P. manshurica* has been used to control postharvest diseases in grapes and citrus fruits [22]. *C. flavescens* was reported to inhibit postharvest gray mold caused by *B. cinerea* in apples, with a reduction rate of 82% and a reduction in decay diameter of *B. cinerea* by more than 60% [23]. The combination of *C. flavescens* and *Bacillus subtilis* reduced the disease symptoms of *Fusarium* head blight caused by *Fusarium graminearum* in wheat [24]. *S. pararoseus* has shown promise as a biocontrol agent against postharvest diseases of grapes and strawberries caused by *Aspergillus niger* and *B. cinerea* by inhibiting the growth of the pathogen and promoting the activities of resistance-related enzymes in the host [25,26]. It was found that the volatile compounds of *S. cerevisiae* were able to inhibit the growth of the pathogenic fungus *B. cinerea* both in vitro and in vivo in strawberries and black spot disease caused by *Phyllosticta citricarpa* in citrus fruits [27,28]. It has been reported that *P. caribbica* shows a strong ability to control apple blue mold and degrade patulin [29]. When treated with glycine betaine or phytic acid, the biocontrol ability of *P. caribbica* against apple blue mold caused by *Penicillium expansum* was improved by lower disease incidence and lesion diameter and higher resistance-related enzyme activities [30,31]. After mass screening, *W. anomalus* was reported to be an exceptional biological control agent that protected tomato from *B. cinerea* infection and reduced disease severity by 97%, and also inhibited soil-borne diseases (*Verticillium dahliae* and *F. oxysporum*) [14]. The *W. anomalus* screened by Zhang et al. also showed strong biocontrol abilities against blue mold in pears, reducing the incidence of the disease to 5.56%, and inducing a relevant enzyme activity that was a maximum of 4.5 times that of the control group [15].

In this research, the biocontrol ability of six different yeast strains, namely, *P. manshurica*, *C. flavescens*, *S. pararoseus*, *S. cerevisiae*, *P. caribbica*, and *W. anomalus* against *B. cinerea* was estimated and compared, and *W. anomalus* proved to be the most exceptional biocontrol agent among them, inhibiting the growth of *B. cinerea* and resulting in the lowest colony diameter, decay incidence, and decay diameter (Figure 1 and Figure 2). It was reported that *W. anomalus* showed a strong ability to control postharvest diseases, such as blue mold in pears and brown rot in apples [15,32]. However, the biocontrol mechanism of *W. anomalus* against tomato gray mold has not yet been discovered and revealed. The results of this study suggest that the more effective biocontrol ability is related to higher yeast concentrations (Figure 3), which is consistent with the research of Zhao et al., who reported that high yeast concentrations reduced decay incidence to less than 20% (10.3%) and lesion diameter to nearly half (8.03 mm) the respective values for the control group (18.35 mm), confirming that increasing the concentration of biocontrol agent plays an important role in enhancing biocontrol effectiveness [33]. To further explore and understand the mechanism of biocontrol of *W. anomalus* against tomato gray mold, several experiments were conducted. Environmental adaptability is one of the essential elements for antagonistic yeast to protect the host from pathogen infection. Yeast with strong environmental adaptability can quickly occupy the space and gain access to nutrition, inhibiting pathogen infection, and reducing the development of decay [34]. In this study, *W. anomalus* colonized tomato wounds rapidly and grew steadily at both 20 °C and 4 °C, but its population remained at lower numbers on the surface of tomatoes at these temperatures, suggesting that tomato wounds provided a suitable environment for *W. anomalus* to develop its ability to colonize and multiply (Figure 4 and Figure 5). Similar results were found by Zhao et al., who reported that the number of *W. anomalus* in apples increased from about 5 × 10^6^ to 2.88 × 10^7^ CFU per wound within 6 days [35]. 

Numerous studies have shown that antagonistic yeasts are able to induce host enzyme activities associated with defense, which plays an important role in postharvest protection of fruits and vegetables from pathogen infections [36,37]. This study showed that the activities of PPO, POD, APX, SOD, and flavonoid and total phenolic contents of tomato were induced by *W. anomalus* and were higher than in the control groups in most treatment periods (Figure 6). Both PPO and POD are involved in the synthesis of some hormones that increase host resistance to diseases [38]. PPO is an oxidative enzyme whose activity is closely associated with fruit tissue browning and systematic resistance of plants to pathogens [39,40]. In plants, PPO is involved in the conversion of phenols to quinones [41], a type of antimicrobial phenolic substances that are toxic to pathogens [17]. This study confirms the research of Zhou et al., who found that the PPO activity of citrus treated with a combination of salicylic acid and *P. membranaefaciens* was 50% higher than that of the control group, which was associated with an improvement in host defenses against pathogen infection [42]. Plant cell viability decreases with the accumulation of reactive oxygen species (ROS), leading to host susceptibility to pathogen infection [43]. Some enzyme activities are related to the regulation of ROS. POD is an antioxidant enzyme that plays an important role in plant defense response [44]. Using various reducing agents as electron acceptors, POD removes peroxides in plants [45], prevents the formation of ROS in cells of vegetables and fruits, and protects living cells from potential damage by ROS [46]. In addition, POD supports the biosynthesis of lignin and phytoalexin, thus strengthening the plant cell wall [47]. Yan et al. reported that the activity of POD was activated by *Meyerozyma guilliermondii* and increased the resistance of pears against blue mold [48], which is consistent with this study. The enzyme APX catalyzes the reaction of ascorbic acid and H_2_O_2_, decreases the H_2_O_2_ content, and increases the reduction potential of the system, which removes the free radicals in cells under the coordination of the antioxidant system [49]. This study is consistent with the findings of Ahima et al., who found that treatment with a combination of *R. mucilaginosa* and salicylic acid leads to an increase in APX activity and strengthening of host resistance to pathogen infection [50]. SOD and POD work together to protect the cell membrane system from damage caused by reactive oxygen species or other peroxides, thereby reducing free radical damage to organisms [51]. SOD is the key enzyme to prevent free radical formation in plants under oxidative stress, which mainly dismutates superoxide anion (O^2−^) free radicals into H_2_O_2_ and O_2_ [52]. Similar results to this study were found in the study of Han et al., who found that the combination of alginate oligosaccharide (AOS) and *M. guilliermondii* increased the SOD activity of pears after harvest by a maximum of 2.5 times that of the control group [53]. In addition to enzymes, some secondary metabolites, such as phenols and flavonoids, are also important in defense against pathogens [37,54]. The content of total phenols and flavonoid compounds is related to the metabolism of L-phenylalanine, which improves the disease resistance of hosts [35]. Some phenolic compounds are cell wall components, and the interaction between phenolics plays an important role in improving host defense systems against pathogens [21]. Overall, postharvest treatment in tomato with *W. anomalus* can induce enzymes that play an important role in plant disease resistance and increase the content of antifungal compounds, such as total phenolics and flavonoids, thereby improving fruit resistance to disease.

## 5. Conclusions

From the current experiment it can be concluded that *W. anomalus* is the best biocontrol agent among the six different potential antagonistic yeasts used to control gray mold disease in tomatoes. However, the biocontrol ability mainly depends on the concentration used. The biocontrol mechanisms mainly include the ability of rapid colonization and steady growth in the host, as well as the promotion of the activity of defense-related enzymes and the content of total phenols and flavonoid compounds in tomato, which ultimately increases the resistance of tomato to the disease. The mechanisms of induced resistance were tested only on the control and yeast without investigating the more complicated interaction between pathogen, yeast, and host. Therefore, further studies are needed to investigate the mechanisms of induced resistance when the pathogen is inoculated and when the pathogen and yeast are inoculated together. The current study will have an important contribution to developing biopesticides as an alternative to chemical fungicides for sustainable agricultural production.

## Figures and Tables

**Figure 1 foods-11-00720-f001:**
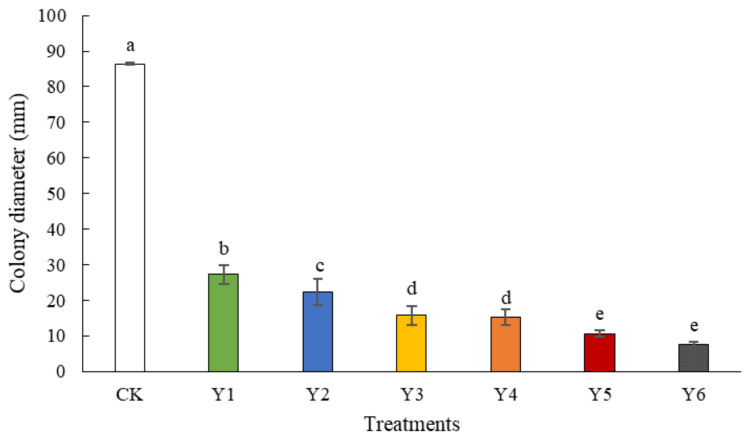
Inhibitory effect of different yeasts against *B. cinerea* on PDA. Colony diameter was collected after 4-day incubation. CK: Tomatoes treated with sterile normal saline. Y1: *P. manshurica*; Y2: *C. flavescens*; Y3: *S. pararoseus*; Y4: *S. cerevisiae*; Y5: *P. caribbica*; Y6: *W. anomalus*. The bars represent standard errors. Different letters indicate a significant difference between treatments according to Duncan’s Multiple Range Test at a value of *p* < 0.05. The experiment was repeated three times (*n* = 3).

**Figure 2 foods-11-00720-f002:**
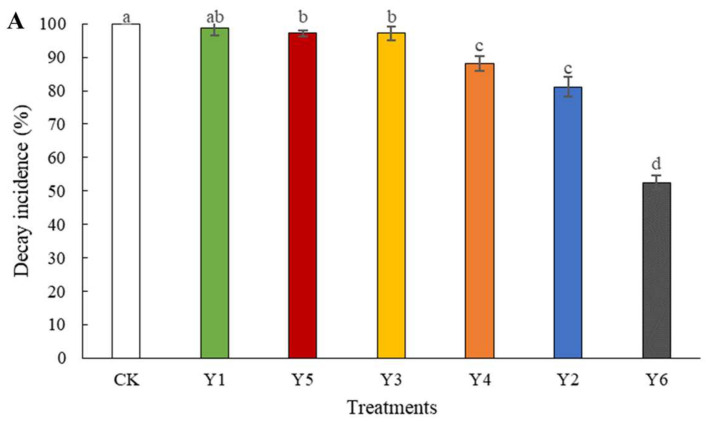
Inhibitory ability of different yeasts against tomato gray mold caused by *B. cinerea*. Decay incidence (**A**) and diameter (**B**) were recorded after 4-day incubation at 20 °C, RH 95%. CK: Tomatoes treated with sterile normal saline. Y1: *P. manshurica*; Y2: *C. flavescens*; Y3: *S. pararoseus*; Y4: *S. cerevisiae*; Y5: *P. caribbica*; Y6: *W. anomalus*. The bars represent standard errors. Different letters indicate a significant difference between treatments according to Duncan’s Multiple Range Test at a value of *p* < 0.05. The experiment was repeated three times (*n* = 3).

**Figure 3 foods-11-00720-f003:**
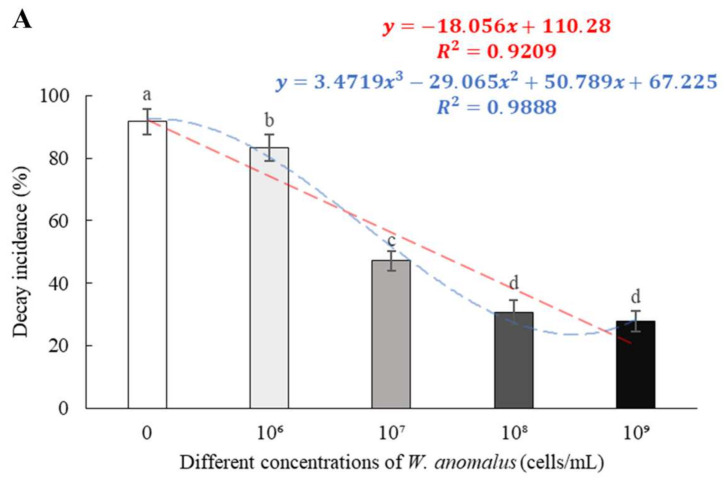
The effect of postharvest treatment of tomatoes with *W. anomalus* at different concentrations on decay incidence (**A**) and lesion diameter (**B**) in tomatoes infected with *B. cinerea* after 4 days of incubation. The bars represent standard errors. Different letters indicate a significant difference between treatments according to Duncan’s Multiple Range Test at a value of *p* < 0.05. The experiment was repeated three times (*n* = 3).

**Figure 4 foods-11-00720-f004:**
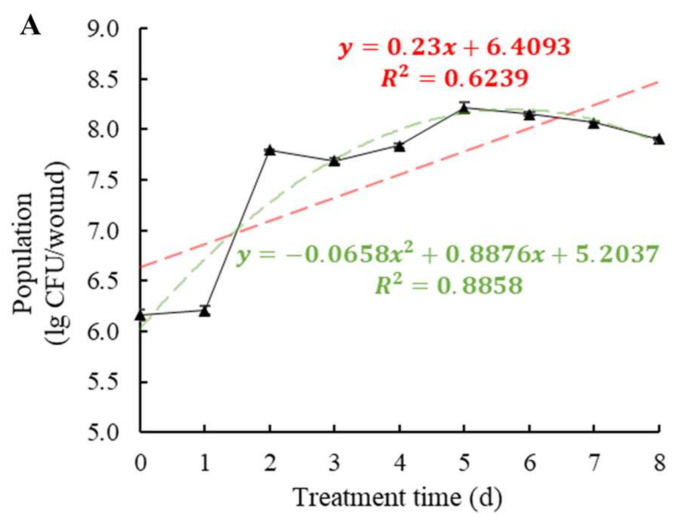
Colonization trends of *W. anomalus* in tomato tissue (**A**) and epidermis (**B**) at 20 ± 2 °C. The bars represent standard errors. The experiment was repeated three times (*n* = 3).

**Figure 5 foods-11-00720-f005:**
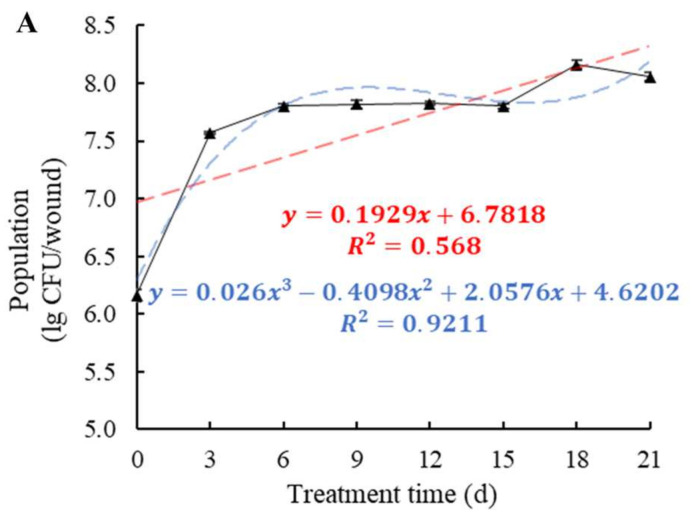
Colonization trends of W. *anomalus* in tomato tissue (**A**) and epidermis (**B**) at 4 ± 2 °C. The bars represent standard errors. The experiment was repeated three times (*n* = 3).

**Figure 6 foods-11-00720-f006:**
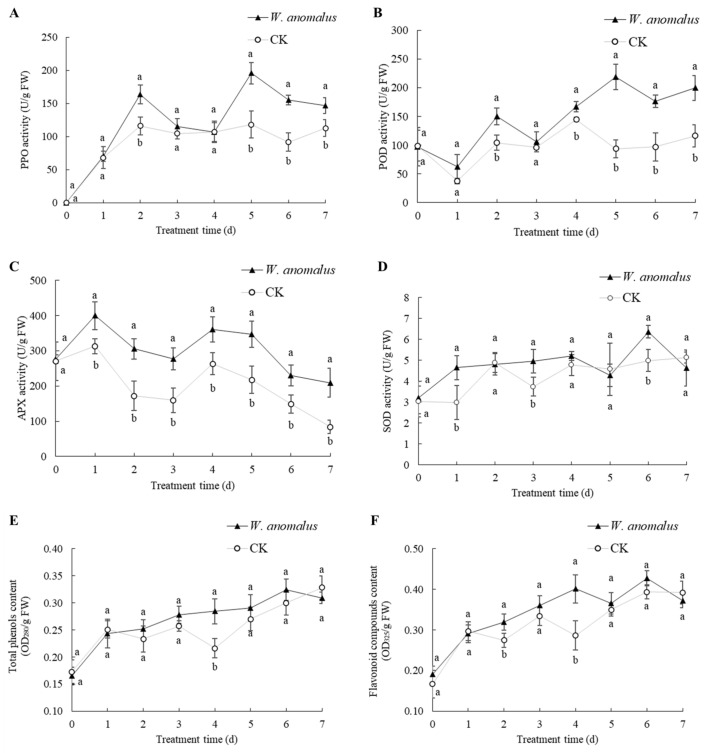
The effects of postharvest treatments of tomatoes with *W. anomalus* on enzymatic activities of polyphenoloxidase (PPO) (**A**), peroxidase (POD) (**B**), aseorbate peroxidase (APX) (**C**), and superoxide dismutase (SOD) (**D**) and contents of total phenols (**E**) and flavonoid compounds (**F**). CK: Control (tomatoes treated with sterile normal saline). The bars represent standard errors. Different letters (a, b) represent significant difference between treatment and control according to Duncan’s Multiple Range Test at a value of *p* < 0.05. The experiment was repeated three times (*n* = 3).

**Table 1 foods-11-00720-t001:** Methods of enzyme extraction and measurement.

Enzyme	Extracting Buffer	Reference
PPO	Phosphate buffer with 1% PVP and 1.33 mM ethylene diamine tetraacetic acid (EDTA)	Apaliya et al. [17]
POD	Phosphate buffer with 1% PVP and 1.33 mM EDTA	Wang, Y. et al. [18]
APX	Phosphate buffer with 0.1 mM EDTA, 1 mM ascorbic acid and 2% polyvinyl-poly-pyrrolidone (PVPP)	Wang, M. et al. [19]
SOD	Phosphate buffer with 5 mM dithiothreitol (DTT) and 5% PVP	Li and Yan [20]

## Data Availability

All data are included in the manuscript.

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
