# Peer review of "Efficacy of the Yeast Wickerhamomyces anomalus in Biocontrol of Gray Mold Decay of Tomatoes and Study of the Mechanisms Involved"

_foods, 2022, doi:10.3390/foods11050720_

Round 1
Reviewer 1 Report
Manuscript "Efficacy of the yeast Wickerhamomyces anomalus in biocontrol of gray mold decay of tomatoes and study of the mechanisms involved" presents the results of studies on the use of yeast in biocontrol against B. cinerea mold. It is an interesting subject of research that can be practically used in industry. The verification of the effect of yeast used for biocontrol and the enzymatic changes resulting from this protection are innovative studies.
Detailed comments:
Abstract - requires editing and providing detailed research results.
Introduction - please expand the paragraph about tomato storage pathogens.
line 77 - what was the required concentration?
Chapter 2.2 - Please describe in more detail the antagonistic yeasts (origin, identification method, place where the strain was deposited)
line 71-72 - please provide reference to previous research.
Chapter 2.3 - How was the pathogen identified?
The results are clearly described, the discussion of the results is correct.
Author Response
Dear Editor,
Manuscript Title: Efficacy of the yeast Wickerhamomyces anomalus in biocontrol of gray mold decay of tomatoes and study of the mechanisms involved. Thank you very much for your suggestion which we also believe will improve the quality of our work. We have revised the manuscript according to your and the reviewers’ comments and would like to resubmit it for your consideration. Based on the reviewers’ comments and suggestions, we have made changes to the original manuscript, which are marked in the manuscript. Below you will find our responses to the reviewers’ comments point by point.
Abstract - requires editing and providing detailed research results.
Response: Dear reviewer, thank you very much for your valuable suggestion. We now modified the abstract and incorporated detailed research results.
Introduction - please expand the paragraph about tomato storage pathogens.
Response: Thank you for the valuable suggestion, we now included relevant content about tomato storage in the revised manuscript. The changes are marked with red color.
Materials and Methods
2.1 Fruits
Line 66 - 69: please specify the tomato variety. Were all the fruits of a similar size? what was their mass, diameter?
Response: Dear reviewer, the tomato variety is now specified and fruit size is now described in the revised manuscript.
2.2 Antagonist
Please describe in more detail the antagonistic yeasts (origin, identification method, place where the strain was deposited)
How was yeast identified? Genetically? Please complete, provide Gen Bank accession numbers.
Line 71 - 72: from what and when were the yeasts originally isolated?
Line 71 - 72: please provide reference to previous research.
Response: Dear reviewer, details of the antagonistic yeasts are now provided incorporated in the revised manuscript. The origin and identification method are described and the place where W. anomalus deposited with accession number and reference to previous research are now provided.
Line 77: what was the required concentration?
Response: Dear reviewer, the used concentrations for the current study are now described in Line 87-88.
2.3 Fungal Pathogen
How was the pathogen identified?
Fungal Pathogen: how was mold identified?
Response: Dear reviewer, thank you for your valuable question here. We missed to incorporate this important information in our manuscript. Now, the detailed identification method is now described in the revised manuscript (Line 92).
Results
The figures are not aesthetic, please change to colored.
Response: Dear reviewer, we now provided colored figures with higher resolution accordingly.

Reviewer 2 Report
66 – 69: please specify the tomato variety. Were all the fruits of a similar size? what was their mass, diameter?
71 – 72: from what and when were the yeasts originally isolated?
2.3. Fungal Pathogen: how was mold identified?
How was yeast identified? Genetically? Please complete, provide Gen Bank accession numbers.
The figures are not aesthetic, please change to colored.
Author Response

(The authors gave the same response as above.)
